# Deriving disease modules from the compressed transcriptional space embedded in a deep autoencoder

Sanjiv K. Dwivedi [1], Andreas Tjärnberg [1,2,3], Jesper Tegnér [4,5,6] & Mika Gustafsson [1✉]

Disease modules in molecular interaction maps have been useful for characterizing diseases. Yet biological networks, that commonly define such modules are incomplete and biased toward some well-studied disease genes. Here we ask whether disease-relevant modules of genes can be discovered without prior knowledge of a biological network, instead training a deep autoencoder from large transcriptional data. We hypothesize that modules could be discovered within the autoencoder representations. We find a statistically significant enrichment of genome-wide association studies (GWAS) relevant genes in the last layer, and to a successively lesser degree in the middle and first layers respectively. In contrast, we find an opposite gradient where a modular protein–protein interaction signal is strongest in the first layer, but then vanishing smoothly deeper in the network. We conclude that a data-driven discovery approach is sufficient to discover groups of disease-related genes.

---

[1] Bioinformatics, Department of Physics, Chemistry and Biology, Linköping University, Linköping, Sweden. [2] Department of Biology, Center For Genomics and Systems Biology, New York University, New York, NY 10008, USA. [3] Center for Developmental Genetics, Department of Biology, New York University, New York, NY, USA. [4] Biological and Environmental Sciences and Engineering Division, Computer, Electrical and Mathematical Sciences and Engineering Division, King Abdullah University of Science and Technology (KAUST), Thuwal 23955–6900, Saudi Arabia. [5] Unit of Computational Medicine, Department of Medicine, Solna, Center for Molecular Medicine, Karolinska Institutet, Stockholm, Sweden. [6] Science for Life Laboratory, Solna, Sweden. ✉email: mika.gustafsson@liu.se

A trend in systems medicine applications is to increasingly utilize the fact that disease genes are functionally related and their corresponding protein products are highly interconnected within networks, thereby forming disease modules[1,2]. Those modules defines systematic grouping of genes based on their interaction, which circumvents part of the previous problems using gene-set enrichment analysis[3] that require pathway derived gene-sets, which are less precise since key disease pathways are highly overlapping[1,4]. Several module-based studies have been performed on different diseases by us and others, defining a disease module paradigm[1,2,5,6]. The modules generally contain many genes and a general principle for validation has been to use genomic concordance, i.e., the module derived from gene expression and protein interactions can be validated by enrichment of disease-associated SNPs from GWAS. The genomic concordance principle was also used in a DREAM challenge that compared different module-based approaches[7]. Yet these studies require as a rule knowledge of protein–protein interaction (PPI) networks to define such modules, which by their nature are incomplete, and either biased toward some well-studied disease genes[8], with a few exceptions[9,10], or derived from simple gene–gene correlation studies.

Deep artificial neural networks (DNNs) are revolutionizing areas such as computer vision, speech recognition, and natural language processing[11], but only recently emerging to have an impact on systems and precision medicine[12]. For example, the performance of the top five error rates for the winners in the international image recognition challenge (ILSVRC) dropped from 20% in 2010 to 5% in 2015 upon the introduction of deep learning using pretrained DNNs that were refined using transfer learning[13]. DNN architectures are hierarchically organized layers including a learning rule with nonlinear transformations[14]. The layers in a deep learning architecture correspond to concepts or features in the learning domain, where higher-level concepts are defined or composed from lower-level ones. Variational autoencoders (VAEs) is one example of a DNN that aims to mimic the input signal using a compressed representation, where principal component analysis represents the simplest form of a shallow linear AE. Given enough data, deep AEs have the advantage of being able both to create relevant features from raw data and identify highly complex nonlinear relationships, such as the famous XOR switch, which is true if only one of its inputs is true (Fig. 1a).

Although omics repositories have increased in size lately, they are still several orders of magnitude smaller compared to image data sets used for ILSVRC. Therefore, effective DNNs should be based on as much omics data as possible, potentially using transfer learning from the prime largest repositories and possibly also incorporating functional hidden node representation using biological knowledge. The LINCS project defined and collected microarrays measuring only ~1000 carefully selected landmark genes, which they used to impute 95% of the remaining genes[15]. Note that this compression can at best work for mild perturbations of a cell for which the DNN has been trained to fit. Hence, they may not generalize well on new knockdown experiments[16].

Although interesting and useful for prediction purposes, those representations in a DNN cannot readily be used for data integration or serve as biological interpretation. For that purpose, Tan et al. used denoising AEs derived from the transcriptomics of *Pseudomonas aeruginosa* and discovered a representation where each node coincided with known biological pathways[17]. Chen et al. used cancer data and showed that starting from pathways represented as a priori defined hidden nodes, allowed the investigators to explain 88% of variance, which in turn produced an interpretable representation[18]. Recently a few authors have shown that unbiased data-driven compression can learn meaningful representations from unlabeled data, which predicted labeled data of single-cells RNA-seq[19,20] and drug responses[21,22]. These results demonstrate that AEs can use predefined functional representations, and can learn such representations from input data that can be used for other purposes in transfer learning approaches.

However, a systematic evaluation of how to balance between predefined features versus purely data-driven learning remains to be determined. To address this question, the interpretation of the representations within NNs is fundamental. The most commonly

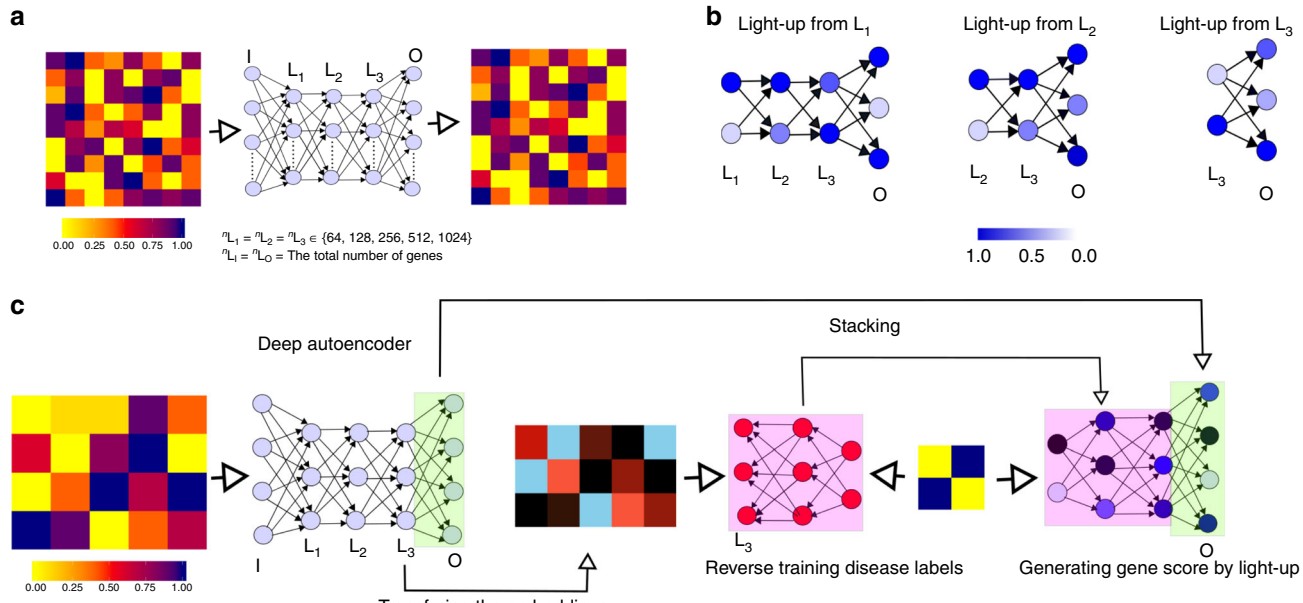

**Fig. 1 Schematic diagram of interpreting an autoencoder and defining the disease modules. a** Training an autoencoder. **b** The steps of light-up method used for interpreting the hidden layer nodes in terms of PPI and pathways. **c** Depicts the steps of predicting the disease gene using transcriptomics signals and autoencoder.

used tool for this purpose is the Google DeepDream[23]. Briefly, the output effect is analyzed using the light-up of an activation of one hidden node, followed by a forward-propagation of the input to the output. This allows the user to interpret the net effect of a certain node and is referred to by us as light-up (Fig. 1b).

In this work, we investigated different AE architectures searching for a minimal representation that explained gene expression, which in our hands resulted in a 512-node wide and three-layered deepAE capturing ~95% of the variance in the data. Next, we derived a novel reverse supervised training-based approach based on light-up of the top of the transferred representations of the trained AE that defined the disease module (Fig. 1c). Using the third layer of the AE we identified disease modules for eight complex diseases and cancers which were all validated by highly significant enrichment of GWAS genes of the same disease. In order to understand the role of each of the hidden representations we tested whether they corresponded to genes that were functionally related and disease associated genes. First, unsupervised analysis of the samples in the AE space showed that disease cluster in all layers, while cell types clustered only in the third layer. Then, we decoded the meaning of the outputs from the nonlinear transformations that defined the compressed space of the autoencoder. To this end we utilized closeness and centrality of the PPI data in STRING[24], as a first step to test if the derived gene-sets was linked to previous disease module research. Conversely, we found that genes within the same hidden AE node in the first layer were highly interconnected in the STRING network, which gradually vanished across the layers. In summary, we believe that our data-driven analysis using deepAE with a subsequent knowledge-based interpretation scheme, enables systems medicine to become sufficiently powerful to allow unbiased identification of complex novel gene-cell type interactions.

## Results

**A deepAE with 512 nodes explained 95% of variance.** Training neural networks requires substantial, well-controlled big data. In this study we therefore performed our analysis using the 27,887 quality-controlled and batch-effect-corrected Affymetrix HG-U133Plus2 array compendium, thus encompassing data from multiple laboratories[25]. Furthermore, the data had previously been analyzed using cluster analysis and linear projection techniques such as principal component analysis[25]. The data set and the ensuing analysis therefore constitute a solid reference point based on which we are in a good position to ask whether successive nonlinear transformations of the data would induce a biologically useful representation(s). Specifically, we investigate whether disease-relevant modules could be discovered by training an autoencoder (AE) using this data set. The underlying hypothesis being that an autoencoder compression represents a nonlinear unbiased representation of the data. Similarly to the knowledge-driven disease module hypothesis[2], closeness within the autoencoder space suggest functional similarity and could be used to identify upstream disease factors.

To this end we partitioned the data into 20,000 training and 7887 test samples. We trained AEs of different widths from 64 to 1024 hidden nodes, incremented stepwise in powers of two, and we contrasted two depths in our analysis, i.e., a single-layered coded shallow AE (shallowAE) and a deep triple layered AE. Depth refers to that the encoder and decoder both contains one extra hidden layer generating two weight sets, respectively, in contrast to the shallow AE which has only a single weight set for encoder and decoder respectively[26]. We calculated the mean squared training and test error (measured using error $= 1 - R^2$, where $R^2$ is computed globally over all genes using a global data variance (Fig. 2a) and locally for each gene individually using gene-wise variances (Fig. 2b, c)). Comparing the reconstruction

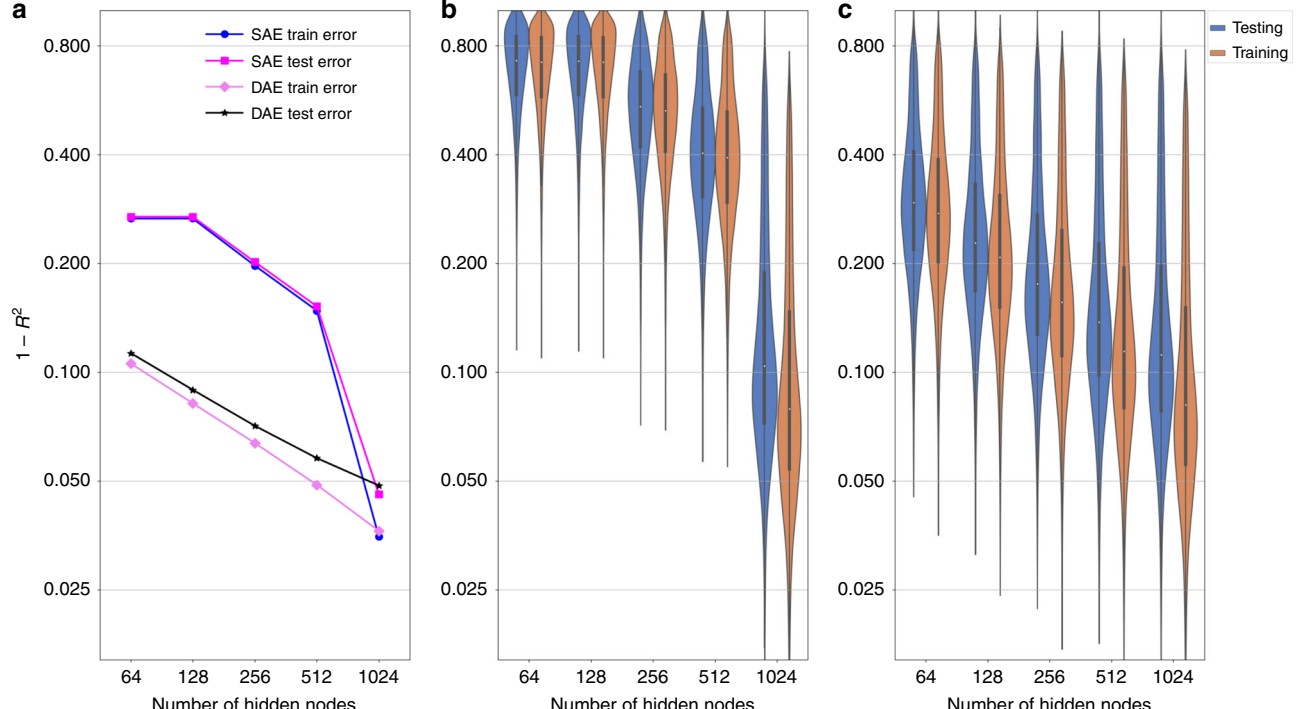

**Fig. 2 Deep autoencoder (deepAE) outperformed shallow autoencoder (shallowAE) up to 512 hidden nodes in terms of accuracy.** $1 -$ coefficient of determination ($R^2$), in training and validation set using the full data set variance (**a**) and the gene-wise variances (**b**, **c**). The left panel shows the mean behavior of $R^2$ values on the full data set. The distribution of $R^2$ values across each gene is shown for both models, shallowAE (**b**), and three-layer deepAE (**c**), with increase in the number of hidden nodes in each layer from 64 to 1024.

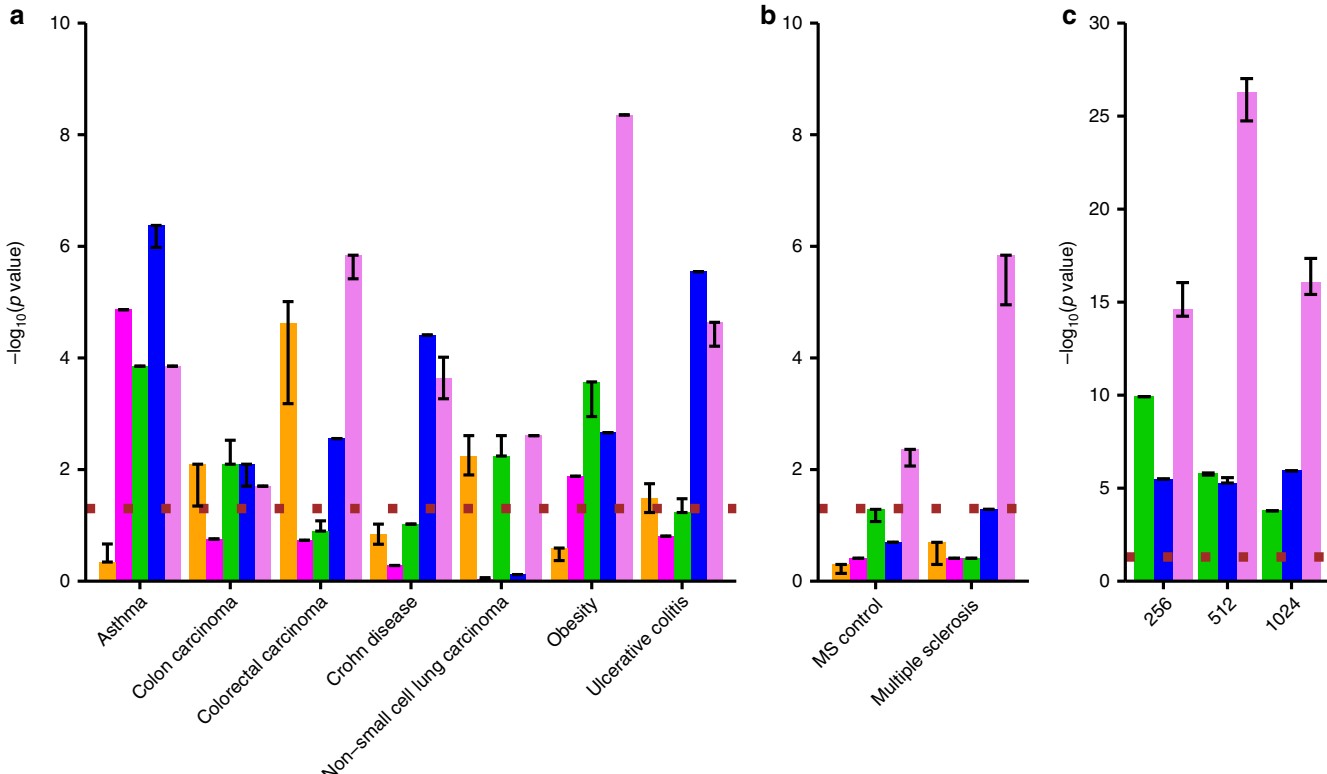

**Fig. 3 Disease association enrichment of autoencoder (AE)-derived gene sets. a, b** Enrichment score ($-\log10(P)$) resulting from the hyper-geometric test between disease gene overlap of the predicted genes by the deep neural network derived by first (green), second (blue), and third (violet) hidden layers of the deep autoencoder (deepAE). As references, we show with a method based on a vanilla supervised neural network (orange) and also the hidden layer of the shallow autoencoder 512 nodes (shallowAE; magenta). MS. **c** The Fisher's combined $p$ value across all eight diseases predicted by a three-layer deep autoencoder. The dotted (brown) line corresponds to the p value, cut-off 0.05.

errors of the different AEs we found, not surprisingly, that the shallow AE performed poorly (>15% error) whenever we used less than 1024 hidden nodes, whereas increasing the number of nodes to 1024, reduced the error threefold to ~5%. In contrast, the deepAE performed well already for 64 hidden nodes (11% error), which subsequently decreased following a power law up to 512 hidden nodes, best described by $R^2 = 0.89 \times 2^{0.028(x - 64)}$, where $x$ is the number of hidden nodes. Next, we analyzed the gene-wise $R^2$ performances of the $R^2$ distributions (Fig. 2b, c), which showed that the median gene error was also low ($R^2 > 0.86$) already for the 512 deepAEs. In summary, we found that the deepAE with 512 hidden nodes performed comparably to the shallowAE with 1024 nodes, although the latter included almost twice as many parameters. Since the purpose of our study was to discover biologically meaningful disease module we proceeded and analyzed the 512 deepAEs in the remaining part of the paper as this architecture provided an effective compression of the data.

**GWAS genes were highly enriched in the third hidden layer.** Our overarching aim was to assess to what extent the compressed expression representation within a deepAE could capture molecular disease-associated signatures in a data-driven manner. To this end we downloaded well-characterized genetic associations for each of the diseases in our data set[27]. From this data we found seven diseases in our gene expression compendium in which at least 100 genes were found in DisGeNET, which we reasoned was sufficiently powerful to perform statistical enrichment analysis. These included asthma, colon carcinoma, colorectal carcinoma, Crohn's disease, nonsmall cell lung cancer, obesity and ulcerative colitis. In order to associate the genes upstream of a disease we designed a procedure which we refer to as reverse training

("Methods"). Briefly, using our hidden node representation and the phenotype vectors (represented as binary coded diseases) we designed a training procedure to predict the gene expression, referred to as 'reverse' since we explicitly used the hidden node representation. This procedure was repeated three times using one hidden layer as input at a time, and as a comparison we also included the shallow AE.

In a result, we deciphered a gene ranking to each disease based on our functional hidden node representation. Next, we assessed the relevance of this representation by computing the overlap of the top 1000 genes using hyper-geometric test for each disease with GWAS (Fig. 3a, b) and as a complementary analysis using disease ontology (Supplementary Fig. 1). Interestingly, we found a highly significant disease association for at least one layer in all tested diseases (Fisher's exact $10^{-8} < P < 0.05$), and for four cases the strongest association was found using the full model. In all the cases, one of the deepAE layers showed higher enrichment than shallowAE. In order to validate the generality of this procedure we downloaded a new data set for MS on the same experimental platform[28]. For this data set we also performed a similar analysis of the control samples with other neurological diseases (OND), similar to the analysis performed in ref. [28]. Reassuringly, we found significant enrichment for MS patients in MS GWAS (Fisher exact test $P = 1.1 \times 10^{-5}$, odds ratio (OR) = 2.4, $n = 30$). Comparing these patients with OND patients showed lower enrichments (Fisher exact test $P = 8.6 \times 10^{-3}$, OR = 1.7, $n = 22$) (Fig. 3b) and a similar amount of top ranked differentially expressed genes between MS and OND showed no significance (Fisher exact test $P = 0.50$, OR = 1.03, $n = 13$). Lastly, to test that the enrichment of GWAS in the third hidden layer was not due to the batch effect and cell-type differences within the compendia we

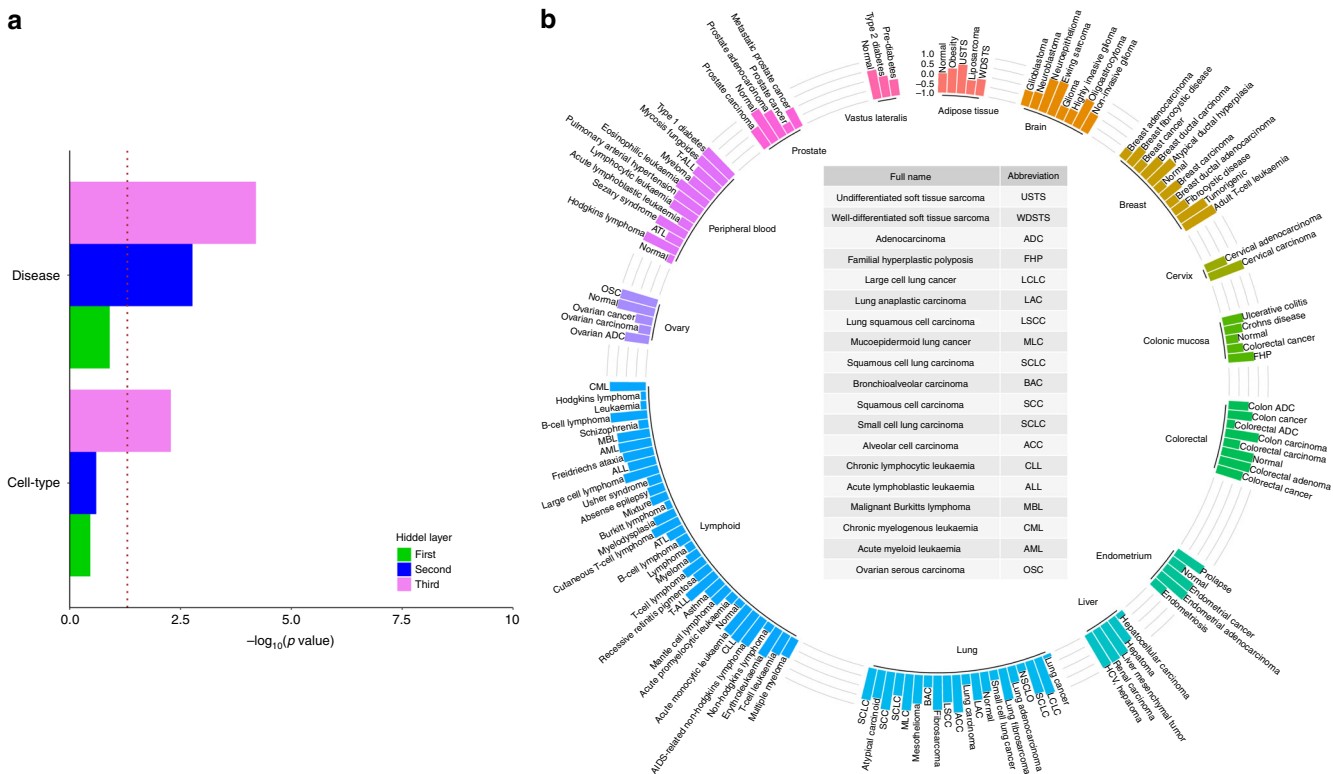

**Fig. 4 Deep autoencoder (deepAE) representation clustering samples into cell types and diseases. a** Significance score ($-\log 10(p)$) for first (green), second (blue), and third (violet) deepAE layers are more coherent (measured by a high Silhouette index (SI)) with respect to cell types (lower) and diseases (upper) than the standard principal component (PC) analysis-based approach. **b** SI defined by the two PCs for diseases and control samples on compressed signals at the third hidden of deepAE with each of the three hidden layers having 512 nodes.

performed similar tests for the control samples. For each study of the eight studied diseases we found higher enrichment than the controls (binomial test $P = 3.5 \times 10^{-2}$, in Supplementary Table 1), supporting the relevance of our disease signatures. Moreover, we compared our unsupervised AE approach followed by reverse training by a naive supervised neural network with 512 hidden nodes ("Methods"). This showed for six out of eight diseases a lower enrichment of GWAS with a tie on the colon carcinoma and nonsmall cell lung cancer. Taken together, the high enrichment of GWAS for the same disease supports our claim that our unbiased nonlinear approach can indeed identify relevant upstream markers, generally with a higher accuracy than the shallower and narrower neural networks.

**Functionally similar samples colocalized in the third layer.** Next, we asked why disease genes preferentially associate with the third but not the other layers in a deepAE. However, to disentangle what is represented by each layer in a deepAE is not straightforward and has previously been the target of other studies[29]. In order to provide insight into what each layer represented in our case, we performed unsupervised clustering of the samples using the compressed representation. Since this was still a 512-dimensional analysis we further visualized the deepAE representation using the first two linear principal components (PCs) of the compressed space. This representation is henceforth referred to as the deepAE-PCA. Previously it has been shown that classical PCA on the full ~20,000-dimensional gene space can discriminate cell types and diseases very well, which we therefore used as a reference in our analysis[25].

To analyze whether samples close in these spaces were biologically more similar than two random samples, we computed the Silhouette index (SI) for phenotypically defined groups,

governed by their cell type and disease status, respectively (Fig. 4). Note that SI = 1 reflects a perfect phenotypic grouping and SI = $-1$ indicates completely mixed samples. Next, the samples were grouped based on the different cell types in the data ($n = 56$) and tested to determine whether the deepAE-PCA or PCA had the highest SI based on each of their respective, different hidden layers (see "Methods"). We filtered the compressed coordinates of normal cell types and found significantly more cell types having a higher SI by at least 0.1 for the third hidden deepAE layer than was the case in the PCA-based approach ($n = 38$ out of 56, odds ratio = 2.11, binomial test $P = 2.28 \times 10^{-3}$). Interestingly, smaller enrichments were also found for the first ($n = 30$, OR = 1.15, binomial test $P = 0.25$) and second ($n = 31$, OR = 1.24, binomial test $P = 0.18$) layers. Next, we repeated this analysis for the 128 diseases in our data-set, and we found again that the third layer ($n = 86$, OR = 2.05, binomial test $P = 6.27 \times 10^{-5}$) exhibited the strongest association with respect to first layer ($n = 71$, OR = 1.25 binomial test $P = 1.25 \times 10^{-1}$) and second layer ($n = 81$, OR = 1.72, binomial test $P = 1.69 \times 10^{-3}$). These observations suggested that samples originating from similar conditions and phenotypes were automatically grouped according to the hidden layers, most significantly for the third.

**First layer associated genes colocalized in the interactome.** In order to further interpret the different layers and uncover their role in defining disease modules, we proceeded to analyze the relationship between the signature genes of each hidden node. Since cellular function is classically described in terms of biological pathways, or lately has also been abstracted to densely interconnected subregions in the interactome (so-called network communities) we analyzed the parameters in the deepAE and their connection to the global pattern of expressed genes[30,31].

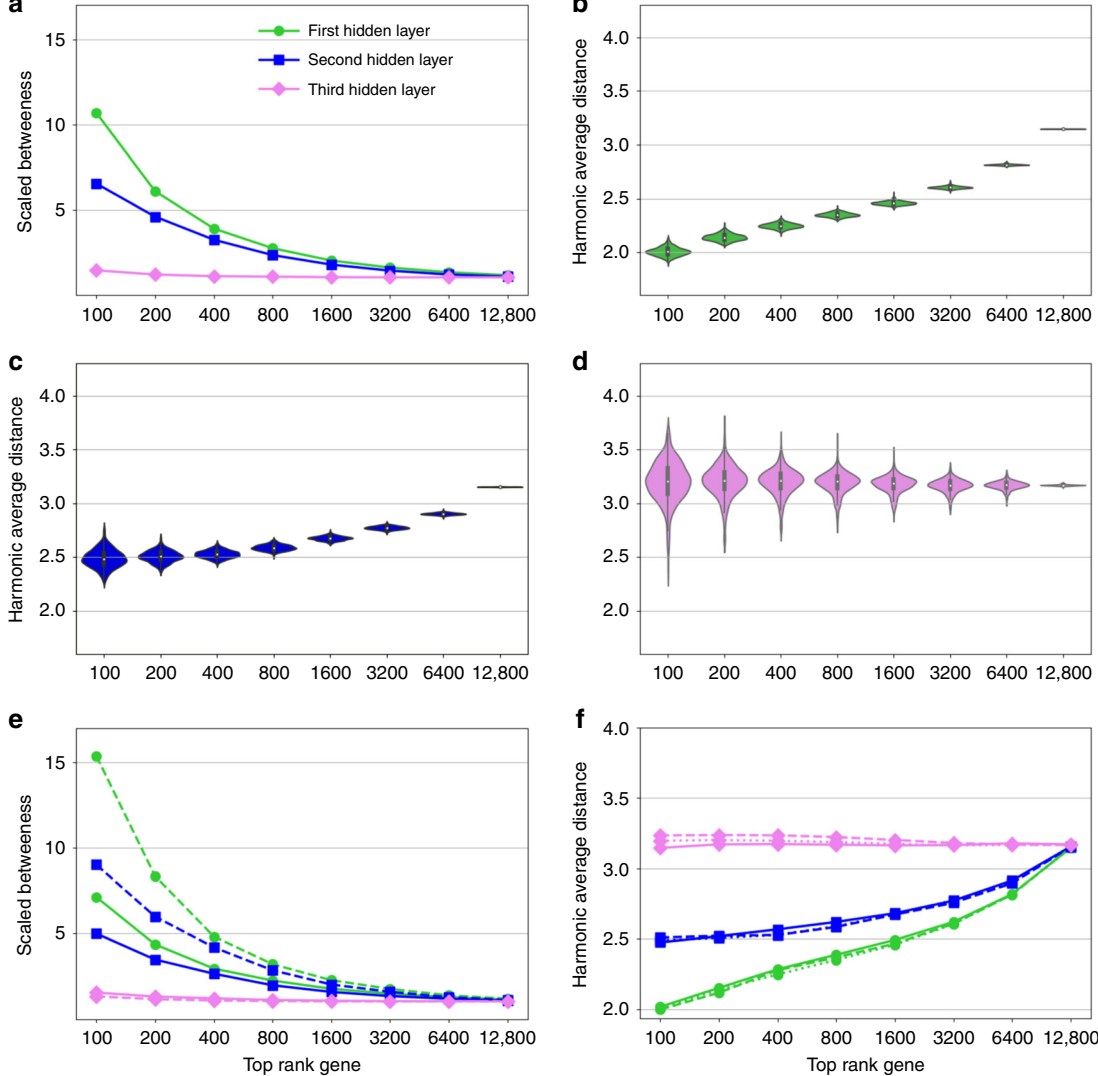

**Fig. 5 Genes that co-localised in the first and seccond hidden layers also co-localised in the interactome. a** The betweenness centrality behavior of the top ranked genes on the basis of the first (green), second (blue), and third (violet) hidden layers of the deep autoencoder. **b–d** The distribution of harmonic average distances of the top rank genes based on each hidden node of the first, second, and third hidden layers of the deep autoencoder, respectively. Also, these results are robust across 256 and 1024 hidden nodes of the deep autoencoder (**e**, **f**).

There are different ways one could potentially interpret parameters in a deepAE. To this end, we created a procedure to associate genes with hidden nodes, which we refer to as light-up. Briefly, a light-up input vector was defined for each hidden node by activating it to the maximum value of the activation function, clamping all other nodes at the same layer deactivated by zero values. Then we forward propagated this input vector through all layers to the output pattern response on the gene expression space ("Methods"). This resulted in a ranked list of genes for each hidden node, identifying which genes were most influenced by the activation of that node. We repeated this procedure for all hidden nodes and layers. In order to test if these lists corresponded to functional units, we analyzed their localization within the PPI network STRING[24]. We hypothesized that genes co-influenced by a hidden node could represent protein patterns involved in the same function. Also, the STRING database captures proteins associated with the same biological function and which are known to be within the same neighborhood of their physical interactome. By first ranking the most influenced genes we systematically analyzed the cutoffs thereby showing whether a gene was considered as associated with the node by powers of two

from 100 to 10,000. Next, we calculated the average shortest path distance between these genes within the STRING network, using the harmonic mean distance to include also disconnected genes.

This analysis revealed that the top-nodes in the ranked lists of the first hidden layer, had a high betweenness centrality (Fig. 5a) while exhibiting a low average graph distance between each other (Fig. 5b–d). Thus, highly co-localized genes were the most central part of the PPI. Both findings were tested using several different cut-offs (Fig. 5a-f), and the effect was most evident for the first layer, appearing to a weaker extent for the second layer and fully vanishing at the third layer. In order to assess the robustness of these results, we investigated the effect of the selection of a specific database and different variants of AE. Specifically, we computed the similarity to our results in the case we used three different annotation databases (BioSNAP, KEGG, REACTOME, and GO, Supplementary Fig. 4 and 5) and compared our approach against deepAE constructed by denoising, and sparse AE as well as funnelings with similar results (Supplementary Figs. 6–9). In all cases we found similar associations across the layers. This analysis suggests that our interpretable gradients in the different layers are robust across these variations.

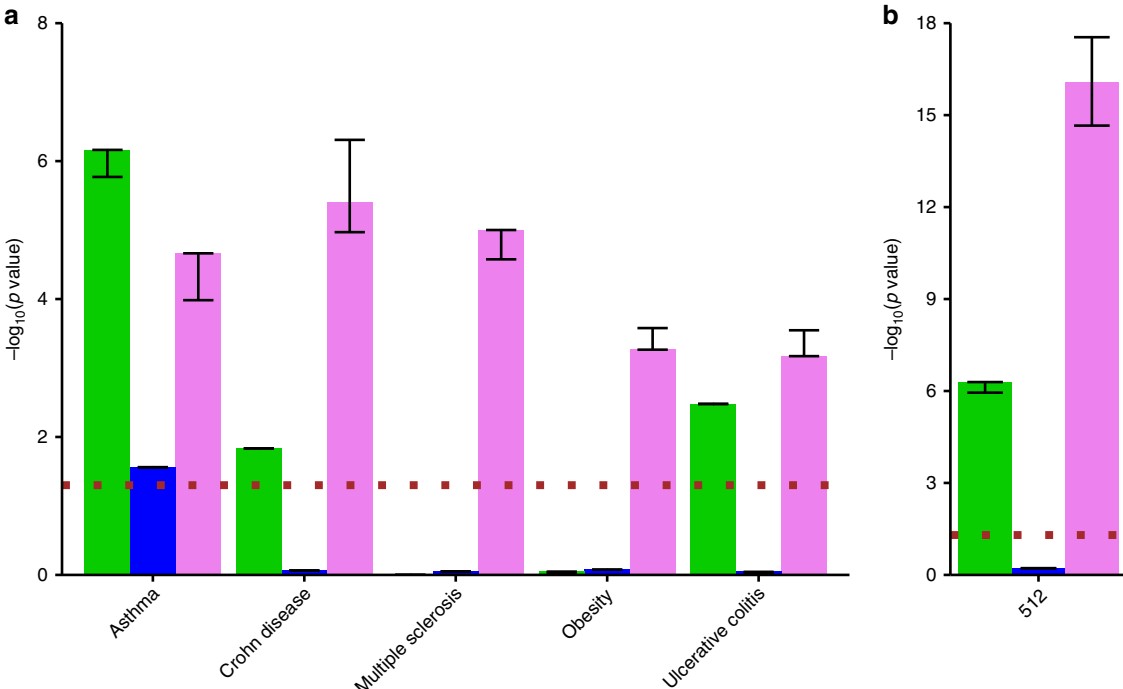

**Fig. 6 Generalization of disease association enrichment results in the deep autoencoder (deepAE) of derived gene sets using RNA-seq data.**
**a** Enrichment score (−log10(P)) resulting from the hyper-geometric test between disease gene overlap of the predicted genes by the deep neural network derived by the first (green), second (blue), and third (violet) hidden layers, of the deepAE. **b** Fisher's combined $p$ value across all five complex diseases predicted by the three-layer deep autoencoder. The dotted (brown) line corresponds to the $p$ value, cut-off 0.05.

**Replications of interpretable gradients using RNA-seq.** In order to assess the generality and to increase the domain of applicability of the AE approach to interpret emerging large RNA-seq data sets, we identified a large publicly available body of RNA-seq material[32]. These data were divided into 50,000 training samples and 9532 validation samples for 18,999 genes, and was used to train a deep AE with similar hyperparameters as for the microarrays, i.e., using a three-layered AE with 512 hidden nodes in each layer. Unfortunately, this data did not contain sufficient complex disease samples, and we therefore searched for additional RNA-seq data sets for our previously tested complex diseases, namely asthma (GSE75011), Crohn's disease, ulcerative colitis (GSE112057), obesity (GSE65540) and multiple sclerosis[33]. Similar to the microarray AE we found a highly consistent significant overlap between GWAS and the associated disease genes derived from the third layer for each of the diseases (Fisher combined $P < 10^{-15}$), and to a lesser extent in the other two layers, see Fig. 6. Next, we tested whether the hidden nodes corresponded to close sets of interconnected protein–protein interactions by repeating the light-up procedure. Interestingly, we found that the top ranked genes in the first, and to a lesser extent also in the second hidden layer, had low average betweenness centrality and had low average distance. Strikingly, this association was even stronger than in the analysis using the AE of the microarrays. In order to understand the reason behind increase in the interpretability level of the PPI association we trained the deep AE on 20 K samples. We found similar association levels as for the 50 K samples (Supplementary Fig. 8). Hence, we conclude that the discrepancies between the microarray and RNA-seq based AEs are not due to the training samples sizes. In summary, our replication of our findings that the relationship between disease gene and the protein interaction confirms our findings of deep AEs as an unbiased estimator of functional disease associations (Fig. 7).

## Discussion

In summary, our study aimed at using deep neural networks for identification of a new unbiased data-driven functional representation that can explain complex diseases without the reliance of the PPI network which is known to be incomplete[7] and strongly affected by the study bias of some early discovered cancer genes. We showed, to the best of our knowledge, for the first time a deep learning analysis do find disease relevant signals and that the different layers capture gradients of biology. This suggests that a data-driven learning approach could eventually complement the findings and techniques derived from network medicine for understanding complex diseases.

In order to find the similar inferences between structural features of the PPI and the estimated parameters of neural networks, we began a systematic demonstration of the light-up concept[23] motivated by the need to prioritize genes based on their contributions in the compressed space of the deepAE. Furthermore, we showed that the top genes prioritized by each node in the first and middle layers are localized and belong to the core part of the PPI. Moreover, the third layer nodes possess long-range variability in showing the localization to delocalization of their top genes compared to the random genes. This kind of gradient in terms of interpretability with respect to localization within the PPI network suggests that each layer indeed encodes different types of biological information. These results also suggest that the transformed signals in the compressed space first decode the modular features of the underlying interactome which then vanishes smoothly layer by layer as a deeper representation is encoded. Concurrently, with such a decreasing protein-defined modular gradient, an increase in disease-relevant genes and modules thereof is progressively discovered in the deepest layers of the AE.

Next, we presented a novel method that uses a supervised neural network to determine a disease-specific feature vector in

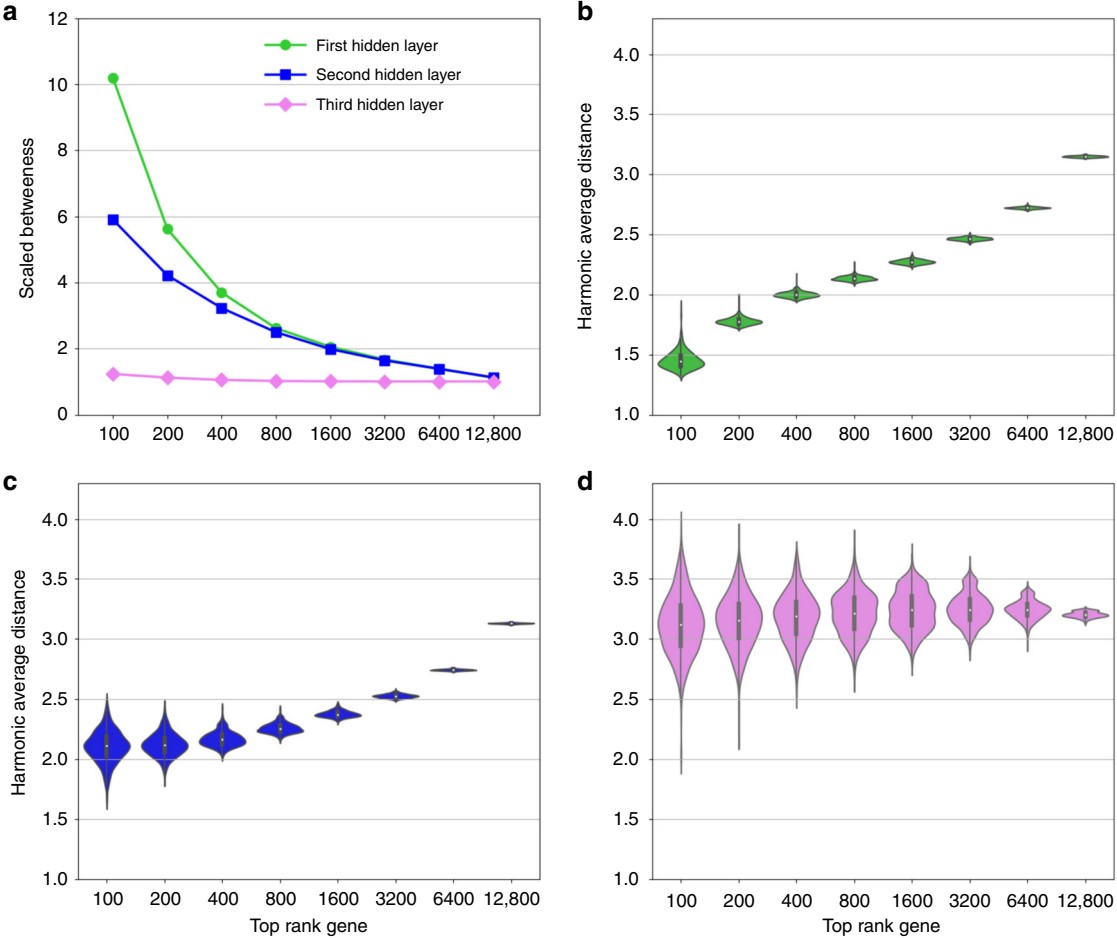

**Fig. 7 RNA-seq replicated gene co-localisation pattern from micro-array data. a** Betweenness centrality behavior of the top ranked genes on the basis of the first (green), second (blue), and third (violet) hidden layers of the deep autoencoder trained on the RNA-seq data. **b–d** Distribution of harmonic average distances of the top rank genes based on each hidden node of the first, second, and third hidden layers of the deep autoencoder respectively.

the compressed space of the deepAE. The disease-specific feature vector of the compressed space was transformed into a gene space that defined the disease module. We compared different AEs and found deepAEs using 512-state variables of ~20,000 genes at a 95% $R^2$ for microarrays and 80–85% for RNA-seq. Interestingly, AE depth reduced the number of learnable parameters approximately twofold from shallow networks and the number of required hidden units, similar to what was reported for a microbiome[17], and also represented a two-fold compression in number of latent variables compared to the number of variables in the LINCS project. The high degree of compression for the deep AE with fewer state variables suggests that this representation is indeed preferable compared to shallow representations. One reason for the need of depth is that such AEs are theoretically capable of capturing more complex relations between genes, such as the XOR relations[26,34,35], which shallow AEs cannot. Importantly, our biological and disease interpretations of the layers were robust w.r.t using microarray or RNAseq data; different data bases for interpretations, as well as different versions of AEs.

Our findings suggest the usefulness of deep learning analysis to decompose different hierarchy levels hidden within the relations between genes. For example, the first layer encodes the modular features belonging to the central part of the interactome. These features are synonymously selected by the interactome-based approaches to find the components that have control over the entire system[36]. In contrast, these features are not necessarily

transferable by cell type-specific transcriptomic signals. Next, the third layer is close to the middle as well as output layer, hence it is proximal in capturing the true cell-type as well as disease specific signals that are coded in terms of interactome. More interestingly, we showed that the third layer efficiently encodes cell type-specific functional features; therefore, it might be reason to increase the likelihood of mapping the disease-specific functional genes by disease-related cell type signals in the light-up. Also, the presented approach can play a crucial role in utilizing the resolution level of the single cell transcriptomic signals in prioritizing genes that are enriched with the upstream dysregulated genes and their relationship with causal genetic variants[37]. Another important application of our approach can indeed provide new insights in the multiscale organization about disease–disease, disease pathways disease–gene associations[38].

An alternative approach to the AE would be using NNs for the particular disease of interest and thereby finding a best representation of the disease. However, although potentially feasible for some diseases, such an approach would in our opinion not likely make best use of the existing compendia for forming latent variables, suggesting that such a representation could fail in generality. Instead, in using transfer learning, our AEs could help stratify disease groups of limited samples as the number of parameters could decrease by about ~40-fold (from ~20,000 to 512), which decrease the analysis complexity. Therefore, transfer could be applied by other clinically interested researchers starting from our derived representation, which could lead to increased

power for building classification systems. We think that the approach is applicable to other omics and using our derived single omics representations together with others they open a door to multi-omics neural networks using transfer learning, similar to what is currently routinely done within the field of image recognition.

## Methods

**Data preparation and normalization.** The available microarray data at Ref. [25] represents normalized log-transformed values. Similarly, we normalized RNA-seq data by the upper quantile method using the function uqua of the R package NOISeq and log transformed the normalized gene expression values by $\log_2(1 +$ normalized expression value)[39]. Also, we discarded the noisy log-transformed expression values those are less than the 3.0. Next, we renormalized both microarray and RNA-seq data such that each ith gene mRNA expression level in the jth sample $E_{i,j}$, across the samples to be in the range between zero and unit,.i.e., $e_{i,j} = \frac{E_{i,j} - \min(E)}{\max(E) - \min(E)}$ .

**Parameter optimization.** The normalized expression matrix $[e_{i,j}]$ is used both as input and output for training the AE with sigmoid activation function. We have chosen a dense layer so that the optimizer starts with an initial point that has unbiased dependency among the data features. We used optimizer ADAM, with learning rate $= 1.0 \times 10^{-4}$, $\beta_1 = 9.0 \times 10^{-1}$, $\beta_2 = 9.99 \times 10^{-1}$, $\varepsilon = 1.0 \times 10^{-8}$ and decay $= 1.0 \times 10^{-6}$, to train the model which we have observed as an optimal choice in predicting the high level of accuracy in both training and validation data sets[40]. The batch size was 256 for the training. In order to systematically investigate the impact of number of hidden nodes on the prediction accuracy, we fixed the number of hidden nodes in all the three hidden layers of the deep AE, termed as a three-layer model (Fig. 1a). In our case, the three-layer model with 512 hidden nodes was more suitable for capturing the biological features. This model has fewer reconstruction errors in comparison with similar hidden node of the one-layer model (shallow AE). Also for the denoising the deepAE, we corrupted the input transcriptomics signals by adding the Gaussian distributed numbers with mean zero and standard deviation 0.5. Next, we replaced the values which are less than zero by zero and greater than 1 by 1. In order to sparsify the deepAE, we used the weight parameter $1.0 \times 10^{-8}$ in the L1 constraint to the kernel regularizer in keras. We implemented our methods using the tensorflow backend (https://www.tensorflow.org) and Keras (https://github.com/keras-team/keras) neural network Python library.

**Interpreting the trained AE with PPI.** The preserved biology in the compressed space is confined in each hidden layer. Therefore, our objective was to understand the meaning of all the nodes in each hidden layer. For this objective, we computed activations at the output layer for each node of a hidden layer. We recursively forward propagated the maximum activation value of each node, while keeping other nodes neutral by zero input, on the remaining portion. Finally, we prioritized the genes on the basis of last layer activations. For simplicity, we mathematically formulated these steps as follows (Fig. 1b). Suppose $k$th layer of an L layer AE, has $N^k$ nodes. Here, $N^1$ and $N^L$ are the same as the number of genes in the profile expression matrix. Also, the number of nodes in each hidden layer is H, i.e., $N^k = H$ for $k \in \{2, 3, 4, \ldots L-1\}$. The following equation recursively defines the activations, $x^k$, of the $k$th layer from the activations, $x^{k-1}$, at $(k-1)$th layer with the initial activation vector $x^p$ (it consists of the maximum activation value at the corresponding position of the hidden node and the rest of the elements are zero) corresponding to the node in the $p$th hidden layer,

$$x^k = \begin{cases} f^k(W^k x^{k-1} + b^k) & \text{if } p < k \le L \\ x^p & \text{if } p = k \end{cases}, \qquad (1)$$

where $f^k$, $b^k$, and $W^k$ are associated with the $k$th layer activation function, bias term and weight matrix respectively. Note that the first input layer does not have an activation function, bias term and weight matrix, so $k \in \{2, 3, 4, \ldots, L\}$. The Eq. (1) defines the activations at the output layer, $x^L$ with dimension of gene size. We prioritized the genes based on the vector, $x^L$, to show the associations with the PPI module.

**Predicting disease genes.** We derived a new approach for predicting a disease gene that is explained in the following four steps (Fig. 1c), which were performed three times in order to estimate mean values and standard deviation estimations:

(1) Compressing the expression profile at hidden layers using trained deepAE.

(2) Training a supervised neural network on the compressed representations in reverse direction: We trained a one-hidden-layer supervised neural network, having the same number of nodes in the second and third layers, with sigmoid and linear activation function respectively. The input matrix $[c_{i,j}]$ is followed by $i \in \{1, 2, 3, \ldots, P\}$ and $j \in \{1, 2, 3, \ldots, S\}$ with dimension $P \times S$, where $P$ and $S$ are the total number of phenotypes and samples respectively. The matrix $[c_{i,j}]$ is defined by another identity matrix $[\delta_{i,p}]$ of the Kronecker tensor as follows, $c_{i,j} = \delta_{ip}$ if the $j$th sample is associated with the $p$th phenotype. The output matrix $[s_{k,j}]$ is a profile

matrix of compressed signals at a hidden layer of dimension $H \times S$, while $H$ is the number of nodes in the hidden layer.

(3) Stacking the supervised neural network with the left part of deepAE, in the feed forward direction, from the layer at which the supervised neural network is trained. We scaled the mean and the variance of the weight matrices and biases in the consecutive layers where both networks are stacked.

(4) Finding the disease scores from the expression: The absolute value of scores $s^p$, for prioritizing the genes related to the $p$th phenotype are computed by the parameters of a stacked neural network using:

$$x^k = \begin{cases} f^k(W^k x^{k-1} + b^k) & \text{if } 1 < k \le L - 1 \\ r^p & \text{if } k = 1 \end{cases},$$

$$s^p = W^L x^{L-1}$$

where $r^p = \left[\delta_{ip}\right]$ is a one column vector for the $p$th phenotype followed by $i \in \{1, 2, 3, \ldots, P\}$.

We compared our approach with naively training, i.e, training gene expression profile as an input instead of the compressed representations, a neural network with 512 hidden nodes and performing disease association as above.

**Validation of predicted genes.** We downloaded the curated disease SNPs from the DisGeNET database and human genome reference consortium assembly, build version 37 (GRch37, hg19) from the UCSC database (https://genome.ucsc.edu/). We computed the closest gene to each disease associated SNP, using Bedtools under the default option. In this way, we defined disease-associated gene sets for validating the neural network-based predicted genes. The performance of the predicted genes was demonstrated in terms of Fisher $p$ value from the hypergeometric test.

**Reporting summary.** Further information on research design is available in the Nature Research Reporting Summary linked to this article.

## Data availability

The trained models and the normalized gene expression data used for defing the disease modules are available at https://figshare.com/articles/Autoencoder_trained_on_transcriptomic_signals/9092045 The microarray and RNA-seq transcriptomics are taken from the ArrayExpress database (accession number E-MTAB-3732) and https://amp.pharm.mssm.edu/archs4/, respectively.

## Code availability

The codes and the tutorial for using them is availiable at the gitlab page https://gitlab.com/Gustafsson-lab/deep_learning_transcriptomics.

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

## Acknowledgements

This work was supported by the Swedish foundation for strategic research, and Swedish Research Council. S.K.D. thanks to Andreas Kalin for helpful discussions and suggestions in deep learning, and Tejaswi VS Badam for his other helpful suggestions.

## Author contributions

M.G., S.K.D. and A.T. conceived the study. S.K.D. performed deep learning training and analysis which were supervised by MG with inputs from A.T. and J.T. All authors contributed in writing and approval of the final draft for publication.

## Competing interests

The authors declare no competing interests.
