## [Peer Review File · Nature Communications]

Reviewers' comments:

Reviewer #1 (Remarks to the Author):

The authors seek to construct a broad model of gene expression in humans by using deep neural networks. There is a lot to like in this paper. It brings together a number of areas of work that have been coexisting but unlinked for a while. One avenue of work has focused on autoencoder-based integration of large data compendia. From a few early papers in 2015 and 2016, the field has come a long-way. However, methods in this domain are often either single-layer and focused on interpretation or multi-layer and focused on modeling the data, perhaps with some transfer step. This manuscript nicely demonstrates that even deep neural networks can be interpreted. There are also a number of biologically interesting findings: protein-protein interactions are captured in lower layers, while groups of disease genes, and presumably biological processes, are captured at higher layers. All of this nicely aligns with expectations of how biological systems are organized. On the one hand, one could argue that there's not a lot new or unexpected here. On the other hand, it nicely unifies work that has been going on across multiple groups with what appears to be a well-designed methodology, and it seems to be an interesting and important contribution in its own right.

The experiments are generally well-designed with reasonable controls. In general, most of my comments are minor as opposed to major. I used major comments to denote things that I felt substantially detracted from the manuscript. I used minor comments to denote things that I felt slightly detracted from the manuscript.

Major:

* The first paragraph of the introduction is a bit much. It would be helpful to focus a little bit more closely on the manuscript at hand. I found the focus on biological networks confusing, since there are already a number of contributions that exist to define co-regulated biological processes and modules via auto encoder nodes. It may be helpful to drop the idea of using networks to define processes entirely, unless it's specifically related to the project at hand. The discussion of UK biobank is also confusing since those assays are a different modality. It feels like this manuscript deserves an intro that's more about this work than generic work.

* Fig 3B should really not be a circle. It's very hard to do anything with those bars.

* There really should be source code, provided under an OSI-approved open license, and input data files associated with this paper that are sufficient for producing the output. The methods section is somewhat terse, so being able to follow the code would be helpful. It would also be good to make the model available.

Minor:

* The authors discuss transfer learning without in gene expression without a reference: "Therefore, effective DNNs should be based on as much omics data as possible, potentially using transfer learning from the prime largest repositories and possibly also incorporating functional hidden node representation using biological knowledge." The authors might consider discussing DeepProfile (<https://www.biorxiv.org/content/10.1101/278739v2>) which is a similar model structure though less focused on transfer, scCoGAPS ([https://www.cell.com/cell-systems/pdfExtended/S2405-4712\(19\)30146-2](https://www.cell.com/cell-systems/pdfExtended/S2405-4712(19)30146-2)) which is a transfer approach using matrix factorization, and/or MultiPLIER ([https://www.cell.com/cell-systems/pdfExtended/S2405-4712\(19\)30119-X](https://www.cell.com/cell-systems/pdfExtended/S2405-4712(19)30119-X)) which is a transfer approach using matrix factorization based on prior biological knowledge.

* The authors discuss a Tan et al. 2016 paper. However, there is a follow-up paper that also explored dimensionality, albeit in a different organism and with different methods. I found it interesting that the dimensionality that parsimoniously models the data is not terribly different than that identified by the 2017 Tan et al. manuscript ([https://www.cell.com/cell-systems/pdfExtended/S2405-4712\(17\)30231-4](https://www.cell.com/cell-systems/pdfExtended/S2405-4712(17)30231-4), especially Figure S2). In some regards this is surprising since one is a microbe and the other is a human dataset, but in other regards perhaps it's somewhat reassuring.

* The LINCS imputation analysis has some issues, and appears to be overly optimistic about the extent to which important variability is captured. The authors state: "The LINCS project defined and collected microarrays measuring only 1000 carefully selected landmark genes, which predicted 95% of the expression of all genes using DNN16." However, others who have looked into predictions of unmeasured genes from LINCS found contradictory results. The most compelling example I have seen is <https://think-lab.github.io/d/185/>. Imputed genes don't seem to vary when knocked down or overexpressed, which calls into question the quality of the imputation. If the authors really want to retain this statement, I'd recommend endorsing the imputation with more caution than they currently do.

* The training compendium is quite small (only 27,887 samples) relative to what is currently available. Other references, including those mentioned in this review, have looked at a broader collection of samples. The analysis of the independent RNA-seq dataset contains more samples. It's interesting to note that the RNA-seq experiment produced some results that suggest that the model there, with more samples, may be more granular. There are some subsampling results in [https://www.cell.com/cell-systems/fulltext/S2405-4712\(19\)30119-X](https://www.cell.com/cell-systems/fulltext/S2405-4712(19)30119-X) that support a model where increasing the number of available samples greatly improves the granularity of discovered processes. It would increase the impact of the paper in terms of our methodological understanding if the authors subsampled the RNA-seq set to match the number of samples from the microarray set, which most of the manuscript focuses on, and reported the extent to which the model changed. The key question: does the RNA-seq model look a little better because it's from a new technology or is it because it has twice as many training samples?

* Missing word: "This represented a two-fold compression compared to the variables in the LINCS project and to our tested shallow."

Reviewer #2 (Remarks to the Author):

The manuscript "Deriving Disease Modules from the Compressed Transcriptional Space Embedded in a Deep Auto-encoder" by Dwivedi et al. describes work on data representations of large-scale gene expression data using deep learning algorithms. Specifically, the paper aims for a better understanding of the relationship of different layers in so-called "auto-encoders" (AE), corresponding to a non-linear data-compression, to gene annotations related to disease phenotypes and molecular networks. AEs are trained by gene expression profiles generated by microarrays probing about 19k genes in a set of 27,887 human samples. The authors compare AEs with different numbers of layers, specifically a "deep" AE (DAE) with three layers and a single layer AE (SAE). They show that DAE work better in compressing gene expression data than SAE with the same number of nodes per layer ranging from 2^6 to 2^9 . Using a concept known as "light-up" they map individual nodes in each layer in DEAs to ranked gene sets in order to relate the layer to gene annotation, either through their annotation to diseases (via GWAS) or their properties in molecular networks (i.e. betweenness and harmonic average distance in protein interaction networks, (PPIs)). They propose a novel "reverse supervised training-based approach" based on "stacking" allowing them to propagate a disease label of the sample to gene scores. The authors also consider RNAseq expression data in what they call "validation".

In a time where "Deep Learning" algorithms are generating considerable success (and hype) in

many domains, including biological data, it is critical to aim for a better understanding of how these methods actually work and how they represent data. In this regard the paper by Dwivedi et al. is very timely and of general interest. Yet, in its current form the manuscript is difficult to read and may also have some conceptual short-comings:

1) While the introduction reads well (except some minor points I mention below), I suspect the average reader of Nature Communications may not have sufficient background knowledge on auto-encoders and their connection to deep learning to easily follow the results section. Much of this information is actually provided in the Methods section (yet not written clearly) and may have been rearranged to follow the format of this journal. The authors should refer to the various boxes before the Results section and rewrite the Methods to more accessible.

2) I missed a discussion of how (unsupervised) data representation (i.e. learning an AE using gene expression data) relates to (disease) prediction using deep networks. If the authors want to claim that their "reverse supervised training-based approach" can profit from the unsupervised data representations, then they should compare the predictive power of their approach to direct training of a deep network on disease states (or at least argue why this is likely to fail and explain that one needs first to dimensionally reduce the feature space to learn a good mapping with the available data).

3) I would like to challenge the authors with regard to the "depth" of the three layers in the DAE: In fact, both the first (L1) and the third layer (L3) are "external" as they connect to the full set of genes, and only the second layer is truly "internal". While L1 clearly is close to the expression data (which it receives as input), this is also true for L3 (as it generates the output). I realise that due to the non-linear activation function the situation is not symmetric, yet the constraint that L3 is proximal to the data layer cannot be ignored.

4) Related to the previous point is the fact that the most conventional organisation of an AE is a funnel, where an 'encoder' first progressively reduces the dimensionality of the data to a 'code' and then a 'decoder' increases the dimensionality back to that of the original data (see scheme at https://en.wikipedia.org/wiki/Autoencoder#/media/File:Autoencoder_structure.png). In contrast, the authors chose all intermediate layers to be of the same size (i.e. use a pipe of equal diameter). I believe the question of whether different depths of the layers in AEs light up gene sets with different properties would better be answered in a funnel AE design (i.e. by comparing the code vs coding or encoding layers).

5) In order to relate a layer to gene annotation, the authors employ gene annotation to diseases via GWAS as a "high-level function" on the one hand and network measures such as betweenness and harmonic average distance in PPIs for "low-level function" on the other hand. This is potentially problematic, since the measures, and not only the biology they relate to, are fundamentally different. An alternative would be to consider the same type of annotation, but only change its nature. Specifically, I suggest contrasting enrichment for GWAS "disease" gene-sets (and preferably also other annotations like those in DGA or OMIM - note that GWAS can only catch genes that have population-wide variants with some but not too big effect sizes to be rapidly purged) with gene-sets of molecular function (e.g. from GO-MF, REACTOME or KEGG pathways). Conversely, the network measures could be contrasted between the PPIs and "disease gene networks" (where genes are linked if they have been associated with the same disease).

6) The authors write as their last sentence in the abstract "We conclude that a data-driven discovery approach, without assuming a particular biological network, is sufficient to discover groups of disease-related genes.", yet they only showed that some of the gene sets that light up are enriched in GWAS disease genes. I believe many data compression methods are likely to find some disease associated modules (and according to Fig. 3A the significance is essentially the same for the three layers, so even PCA should catch some "disease modules" from large-scale expression data). The question is rather how many and how accurate are they, and how do they

compare across methods (including PCA/SAE)? Also, they should compare enrichment with that of other disease modules (e.g. those identified by the DREAM challenge of Ref. 10]). Ideally, to compare their usefulness, one should study their predictive value in disease classification (e.g. using ROC analysis).

7) The RNAseq expression data analysis appears disconnected, and there are few significant associations in Fig. 5. Validation would mean that DAE trained on microarray data, give disease enrichments on RNAseq data, and vice-versa. This would demonstrate that the AE didn't learn artifacts related to the experimental techniques, but rather real biological (gene) signatures.

Besides these major points, I also have some less critical remarks:

1) Sparse AE (SAE) and Denoising AE (DAE) have been shown to work better than simple AE, so I recommend trying those as well. (Also note that their abbreviations unfortunately overlap with those used by the authors for shallow and deep AE).

2) In the abstract the authors write "Yet biological networks, commonly used to define such modules are incomplete and biased toward[s] some well-studied disease genes": I think this depends very much on the network. For example, co-expression networks, are hardly biased towards well-studied genes, as RNAseq (and even modern microarrays) cover essentially all transcripts. Even for PPI there are (raw) experimental data with little bias, if any, but of course this is different when looking at integrated network, such as STRING, that may use literature searches etc. to complement and enforce links with additional (human biased) annotation.

3) 3B\$ as a lower limit for new drug might be a bit much. <https://www.policymed.com/2014/12/a-tough-road-cost-to-develop-one-new-drug-is-26-billion-approval-rate-for-drugs-entering-clinical-de.html> quotes the out-of-pocket cost of \$1.4 billion, but also notes the time cost (estimated at \$1.2 billion), so it's not far off ... Maybe give more than one reference and avoid an exact figure?

4) "Moreover, individual markers for drug selection generally work poorly, and the choice of drugs in complex diseases is often based on trial and error strategies, causing suffering for patients and increasing costs for health care.": Note that some individual markers work quite well: genotyping of the CYP2D6 enzyme is predictive of the processing of many antidepressants and antipsychotic medications, and genotypes in IL28B strongly associate with treatment success of chronic hepatitis C. Also some cancer therapies strongly rely on genotyping nowadays.

5) "Omics could potentially revolutionize medicine by analyzing disease": I'd use 'omics' as a short of '-omics data', but not for '-omics data analysis'.

6) "Systems medicine applications to date have often utilized the fact that disease genes are functionally related and their corresponding protein products are highly interconnected and co-localized within networks []": I think this can only be described as a trend. Also, what's colocalization here (other than being highly connected)?

7) "To this end we utilized the protein-protein interaction data in STRING, as a first step to remove the essence of the knowledge of interactome in defining the phenotypic modules.": Essence of knowledge? Not clear.

8) "The rationale is that by inducing identity mapping from input to output we can readily inspect the resulting deep representation from a disease module standpoint.": Reformulate.

9) Fig. 1A: First the DAE has 3x as many parameters as the SAE, so the comparison is not straightforward. Still, 3x64 and 3x128 DAE do better than a 512 SAE, so I'd point that out. Better would be to have SAEs with 192, 384, ... nodes. Finally SAE overtake DAE between 512 and 1024 nodes, yet it would be good to know more accurately at what node number.

10) Fig. 2: More space between A and B!

11) Fig. 3: Are the bars representing averages (there is one light-up for each node)? If so, also show the stand deviation (or full distribution of p-values). As there are 512 light-ups giving rise to gene sets, shouldn't there be a correction for multiple hypotheses testing? Also, the light-up of each node results in a set of gene scores at the output layer. What cutoff is used to define discrete gene sets (allowing to users Fisher's exact test)? How about using classical gene enrichment using the score itself (e.g. by testing if the GWAS disease genes tend to fetch higher scores than expected)? Also, make color code for layers consistent with Fig 2.

12) Methods: "The micro-array data is log transformed normalized values.": Did you mean "The microarray data were first normalized by a log transformation."?

13) "The normalized expression matrix [...] is input and output signals for training the auto-encoder": Did you mean "The normalized expression matrix [...] is used both as input and output for training the auto-encoder"?

14) "This model has fewer reconstruction errors in comparison with similar hidden node of the one-layer model (SAE)": Rephrase. (Also see my previous comment on comparing with the same number of nodes per layer, but different total number of nodes.)

15) The text following "(2) Training a supervised neural network" is unclear and seems to have mistakes.

16) "The performance of the predicted genes was demonstrated in terms of Fisher p-value using a hypergeometric test.": Strictly speaking Fisher's (exact) test is different from a hypergeometric test: The former converges to the latter for large numbers, but also works exactly for small numbers.

Referee 1:

Major comments:

Comment 1.1: *“The first paragraph of the introduction is a bit much. It would be helpful to focus a little bit more closely on the manuscript at hand. I found the focus on biological networks confusing, since there are already a number of contributions that exist to define co-regulated biological processes and modules via auto encoder nodes. It may be helpful to drop the idea of using networks to define processes entirely, unless it’s specifically related to the project at hand. The discussion of UK biobank is also confusing since those assays are a different modality. It feels like this manuscript deserves an intro that’s more about this work than generic work.”*

Reply 1.1: We agree with the reviewer comment that the previous introduction was too general. We have now removed the introductory sentences regarding molecular networks for precision medicine and UK Biobank. In addition, we clarified the auto-encoder part of the introduction.

Comment 1.2: *“Fig 3B should really not be a circle. It’s very hard to do anything with those bars.”*

Reply 1.2: We agree that our previous figure was hard to interpret as the disease labels were not present. We have now added disease labelling to the plot so the clustering per disease and cell type could be analysed. We revised Fig 3 B as Fig 4 b in the revised manuscript.

Comment 1.3: *“There really should be source code, provided under an OSI-approved open license, and input data files associated with this paper that are sufficient for producing the output. The methods section is somewhat terse, so being able to follow the code would be helpful. It would also be good to make the model available.”*

Reply 1.3: We agree with the reviewer about this and our full OSI-approved open license code is freely available at https://gitlab.com/Gustafsson-lab/deep_learning_transcriptomics. Also, we upload the important models and files on figshare https://figshare.com/articles/Autoencoder_trained_on_transcriptomic_signals/9092045 and linking it to the gitlab page https://gitlab.com/Gustafsson-lab/deep_learning_transcriptomics/blob/master/README .

We have linked the source code and related files in the revised manuscript before the Reference section as follows:

“Data availability

The trained models and the files of the gene expression used in reverse training are available at https://figshare.com/articles/Autoencoder_trained_on_transcriptomic_signals/9092045

The microarray and RNA-seq transcriptomics are taken from the ArrayExpress database (accession number E-MTAB-3732) and <https://amp.pharm.mssm.edu/archs4/> respectively.

Code availability

The codes and the tutorial for using them is available at the gitlab page https://gitlab.com/Gustafsson-lab/deep_learning_transcriptomics. “

Minor comments:

Comment 1.4: “The authors discuss transfer learning without in gene expression without a reference: “Therefore, effective DNNs should be based on as much omics data as possible, potentially using transfer learning from the prime largest repositories and possibly also incorporating functional hidden node representation using biological knowledge.” The authors might consider discussing DeepProfile (<https://www.biorxiv.org/content/10.1101/278739v2>) which is a similar model structure though less focused on transfer, scCoGAPS ([https://www.cell.com/cell-systems/pdfExtended/S2405-4712\(19\)30146-2](https://www.cell.com/cell-systems/pdfExtended/S2405-4712(19)30146-2)) which is a transfer approach using matrix factorization, and/or MultiPLIER ([https://www.cell.com/cell-systems/pdfExtended/S2405-4712\(19\)30119-X](https://www.cell.com/cell-systems/pdfExtended/S2405-4712(19)30119-X)) which is a transfer approach using matrix factorization based on prior biological knowledge.”

Reply (1.4): We thank the reviewer for including these recent works on decomposition of expression data and transfer learning, which we have now added to the introduction.

Comment 1.5: “The authors discuss a Tan et al. 2016 paper. However, there is a follow-up paper that also explored dimensionality, albeit in a different organism and with different methods. I found it interesting that the dimensionality that parsimoniously models the data is not terribly different than that identified by the 2017 Tan et al. manuscript ([https://www.cell.com/cell-systems/pdfExtended/S2405-4712\(17\)30231-4](https://www.cell.com/cell-systems/pdfExtended/S2405-4712(17)30231-4) , especially Figure S2). In some regards this is surprising since one is a microbe and the other is a human dataset, but in other regards perhaps it’s somewhat reassuring.”

Reply (1.5): We thank the referee for this striking observation which we have included in the revised discussion. Also we have also added citations to the Tan 2017 paper in the introduction and discussion.

Comment 1.6: “The LINCS imputation analysis has some issues, and appears to be overly optimistic about the extent to which important variability is captured. The authors state: “The LINCS project defined and collected microarrays measuring only 1000 carefully selected landmark genes, which predicted 95% of the expression of all genes using DNN16.” However, others who have looked into predictions of unmeasured genes from LINCS found contradictory results. The most compelling example I have seen is <https://think-lab.github.io/d/185/> . Imputed genes don’t seem to vary when knocked down or overexpressed, which calls into question the quality of the imputation. If the authors really want to retain this statement, I’d recommend endorsing the imputation with more caution than they currently do.”

Reply 1.6: We thank the reviewer to bringing this github page up. We have added a notion on the interpretation of the imputations related to knock-down experiments in the introduction, reading: “Note that this compression can at best work for mild perturbations of a cell for which the DNN has been trained to fit and as been seen by others might not generalise to new type of knock-down experiments [<https://think-lab.github.io/d/185/>]”.

Comment 1.7: “The training compendium is quite small (only 27,887 samples) relative to what is currently available. Other references, including those mentioned in this review, have looked at a broader collection of samples. The analysis of the independent RNA-seq dataset contains more samples. It’s interesting to note that the RNA-seq experiment produced some results that suggest that the model there, with more samples, may be more granular. There are some subsampling results in [https://www.cell.com/cell-systems/fulltext/S2405-4712\(19\)30119-X](https://www.cell.com/cell-systems/fulltext/S2405-4712(19)30119-X) that support a model where increasing the number of available samples greatly improves the granularity of discovered processes. It would increase the impact of the paper in terms of our methodological understanding if the authors subsampled the RNA-seq set to match the number of samples from the microarray set, which most of the manuscript focuses on, and reported the extent to which the model changed. The

key question: does the RNA-seq model look a little better because it's from a new technology or is it because it has twice as many training samples?"

Reply 1.7: We agree with the reviewer that the size of training compendia is of great importance for deep learning and could indeed also affect the granularity of different AEs. As suggested by the reviewer we therefore trained the auto-encoder using three subsampling experiments with the same sample size as the microarray (i.e. 20,000 training samples). We then analysed the level of PPI association using betweenness centrality and average distance between the genes associated with the hidden nodes, similar to what we did for the fully trained auto-encoder. We found that these metrics closely match the ones for the deep AE trained on all data. Thus, we conclude that the existing quantitative differences between the microarray and the RNA-seq derived ones is likely due to sensitivity and coverage of the different measurement techniques. This additional analysis is shown in Fig. S8 also discussed upon this results in the revised discussion.

Comment 1.8: *"Missing word: "This represented a two-fold compression compared to the variables in the LINCS project and to our tested shallow.""*

Reply 1.8: We have now clarified this sentence and the sub-sentence now reads *"...and also represented a two-fold compression in number of latent variables compared to the number of variables in the LINCS project."*

Referee 2:

Major comments:

Comment 2.1: *"While the introduction reads well (except some minor points I mention below), I suspect the average reader of Nature Communications may not have sufficient background knowledge on auto-encoders and their connection to deep learning to easily follow the results section. Much of this information is actually provided in the Methods section (yet not written clearly) and may have been rearranged to follow the format of this journal. The authors should refer to the various boxes before the Results section and rewrite the Methods to more accessible."*

Reply 2.1: We thank the reviewer for helping us to increase the understanding of the paper. As suggested by the reviewer we have moved them to the introduction and also simplified them. This enabled a higher level of understanding by the reader already in the introduction, which we feel improved the readability of the manuscript.

Comment 2.2: *"I missed a discussion of how (unsupervised) data representation (i.e. learning an AE using gene expression data) relates to (disease) prediction using deep networks. If the authors want to claim that their "reverse supervised training-based approach" can profit from the unsupervised data representations, then they should compare the predictive power of their approach to direct training of a deep network on disease states (or at least argue why this is likely to fail and explain that one needs first to dimensionally reduce the feature space to learn a good mapping with the available data)."*

Reply 2.2 We agree that training a deep neural network directly for predicting disease would be an alternative more direct approach for the specific purpose of disease classification, which might work for disease classification in situations where enough data exists. However, the nature of complex diseases is not rarely relying on rather modest sample numbers. Our work showed that even in those situations our deep AE approach to work well being highly translatable to multiple

situations such as disease, cell-type classifications, protein-protein gene set associations and identification of upstream disease associated genes. We believe that the reason for this success is that the identification of relevant features in the data could be performed using the full data using more than 20,000 samples and that this derived unbiased latent representation is a functionally relevant description of the data. As suggested by the reviewer, we also performed direct training of a three layered deep neural network for disease classification. We first observed that the predictive power in terms of phenotypic classification is very high for both our approach and the direct approach. Secondly, the pathological relevance of the underlying representations differ substantially. The deep AE approach had significantly higher GWAS disease enrichment for seven out of eight tested diseases with no difference in the last diseases. This pinpoints that there are multitude ways to discriminate disease, but our approach first grouping similar genes followed by classification is more likely to generate functionally relevant gene sets. We added this information in the revised Fig. 2 as Fig. 3 in the revised manuscript and discussed upon this in the manuscript.

Comment 2.3: *“I would like to challenge the authors with regard to the “depth” of the three layers in the DAE: In fact, both the first (L1) and the third layer (L3) are “external” as they connect to the full set of genes, and only the second layer is truly “internal”. While L1 clearly is close to the expression data (which it receives as input), this is also true for L3 (as it generates the output). I realise that due to the non-linear activation function the situation is not symmetric, yet the constraint that L3 is proximal to the data layer cannot be”*

Reply 2.3: We agree that the depth of AEs is an important attribute and it is important to have the right notion of depth, and different authors use different definitions on depth. To quote leading authorities in the field *“there is no single correct value for the depth of an architecture”* (Goodfellow, Bengio & Courville *Deep learning* MIT Press, page 8). For a discussion about the definition of deep AE see Goodfellow et al page 499 – 500, which we here try to follow, i.e. we would like to avoid identifying layers as internal or external. If we include the number of input and output layers according to the reviewer wikipedia link we have five layers (L-input, L1, L2, L3, L-output) fully forward connected through four interacting weight matrices (W1,W2,W3,W4). Then (L-input, L1, L2) consists of the encoder and (L2,L3, L-output) the decoder. Each of these contain two weight matrices (W1,W2) and (W3,W4) respectively. Thus we like to keep our previous definition, but has clarified this in the revised results when we introduce the different types of auto-encoders. Regarding the last part of the comment we agree that both L1 and L3 are proximal to the expression data, although the nonlinearity functions makes the mapping non-trivial. However, the auto-encoder comprises of a series of four adjacent nonlinear mappings, so the similarity argument could be used for any of the layers, yet they are different. This also agrees with our findings that gradually changes across the layers, with the most extreme values being in the outer layers.

Comment 2.4: *“Related to the previous point is the fact that the most conventional organisation of an AE is a funnel, where an ‘encoder’ first progressively reduces the dimensionality of the data to a ‘code’ and then a ‘decoder’ increases the dimensionality back to that of the original data (see scheme at https://en.wikipedia.org/wiki/Autoencoder#/media/File:Autoencoder_structure.png). In contrast, the authors chose all intermediate layers to be of the same size (i.e. use a pipe of equal diameter). I believe the question of whether different depths of the layers in AEs light up gene sets with different properties would better be answered in a funnel AE design (i.e. by comparing the code vs coding or encoding layers).”*

Reply 2.4:

Certainly, there are many ways to construct deep AE and funneling is indeed a also standard approach. However, as this also introduces two extra hyper-parameters (namely the depth and

incremental compression) that might be also be tuned we originally used the pipe approach that we felt simplest. However, as suggested by the reviewer we have also included two simple thunnel approaches in our revised supplementary data (Fig. S9), namely (a) a compressed middle layer of 128 nodes (i.e. 512-12-512) and (b) a three layered intermediate funnel making the AE 512-256-128-256-512. Analysing the associations of these layers shows similar results as our original pipe approach with the intermediate layers 2 and 4 are close to their neighboring layers. Indeed, there are many more ways to create funnelling AEs, but as the results of these are similar to the pipe approach we feel confident about the generality of our findings.

Comment 2.5: “In order to relate a layer to gene annotation, the authors employ gene annotation to diseases via GWAS as a “high-level function” on the one hand and network measures such as betweenness and harmonic average distance in PPIs for “low-level function” on the other hand. This is potentially problematic, since the measures, and not only the biology they relate to, are fundamentally different. An alternative would be to consider the same type of annotation, but only change its nature. Specifically, I suggest contrasting enrichment for GWAS “disease” gene-sets (and preferably also other annotations like those in DGA or OMIM - note that GWAS can only catch genes that have population-wide variants with some but not too big effect sizes to be rapidly purged) with gene-sets of molecular function (e.g. from GO-MF, REACTOME or KEGG pathways). Conversely, the network measures could be contrasted between the PPIs and “disease gene networks” (where genes are linked if they have been associated with the same disease).”

Reply 2.5: We agree that the network measures used are different, but they both are fundamental for the disease module hypothesis, which has been defined by Menche et al in Science 2015 (PMID:

25700523) and used by multiple systems medicine researchers (including the authors of this manuscript, see for example our articles (PMIDs:24401939, 25473422, 27626663, 31358043). This hypothesis claims that complex diseases can be seen as malfunctioning of disease modules (Menche et al, Science 2015), and that such malfunctioning can be validated by genomic concordance. Genomic concordance has been defined by us with meaning (PMID:25473422) that a module defined by for example transcriptomics should also be significantly enriched for genes associated with independent omics, for example GWAS. The hypothesis further states that the disease modules should be enriched for genes highly interconnected in the PPI network. In this manuscript we explored how our deepAE latent representation could encapsulate the essence of disease modules. Having said this, we also performed the complementary analyses suggested by the reviewer to further strengthen our claims. First, we tested that the genes associated with the same hidden node was also associated with the same annotation with respect to GO-MF, REACTOME and KEGG pathways. Indeed, as suggested by the PPI association we found annotation association to be strongest for the first layer and to a gradually decreasing extent for second and third layers respectively. We feel that this complementary analysis support our claims, although we feel that as pathways do highly overlap and are divided somewhat arbitrary. Therefore, we believe our PPI approach to be more powerful as it instead focus on the underlying network topology. Second, we recomputed enrichments of the reverse training approach using disease ontology (as we found too few OMIM genes and found DGA less updated). We again found similar enrichments in the third layer according to our original findings. Third, as suggested by the reviewer, we also used a gene-network where interactions are present if two genes are involved in the same disease. Again, although this network has a different meaning we found the strongest association in the first hidden layer. In summary, we believe that all these extra analyses strengthened these claims and have included them in our online supplement and commented upon in the manuscript.

Comment 2.6: “The authors write as their last sentence in the abstract “We conclude that a data-

driven discovery approach, without assuming a particular biological network, is sufficient to discover groups of disease-related genes.”, yet they only showed that some of the gene sets that light up are enriched in GWAS disease genes. I believe many data compression methods are likely to find some disease associated modules (and according to Fig. 3A the significance is essentially the same for the three layers, so even PCA should catch some “disease modules” from large-scale expression data). The question is rather how many and how accurate are they, and how do they compare across methods (including PCA/SAE)? Also, they should compare enrichment with that of other disease modules (e.g. those identified by the DREAM challenge of Ref. 10)]. Ideally, to compare their usefulness, one should study their predictive value in disease classification (e.g. using ROC analysis)”

Reply 2.6:

We agree with the reviewers concerns about more control experiments to justify this claim more properly and has therefore added this at several places as well as clarified some parts of the analysis to avoid possible confusion. We have now added an extra introductory figure which has changed the figure numberings and in our response we use the new labelling, which means that Fig. 3A now is Fig 4a and so on. First, our GWAS analysis from the reverse training using the auto-encoder compressed representation of expression data indicated that our procedure identified genes associated on the DNA level thereby being upstream in the disease. We initially compared our approach by a shallow AE (which is rather similar to a PCA) association and found our approach to have significantly higher enrichment for all diseases except asthma. However, as suggested by the reviewer alternative approaches indeed exists. We agree with the reviewer that disease module based approaches with the specific aim of identifying disease genes would suffice this, which the authors of this article has been working extensively with (PMIDs:25473422, 24571673, 24401939, 27626663, 31010371, 31358043). However, such methods implicitly assumes an underlying network which in most cases suffer from incompleteness and knowledge-bias problems of the interactome. An important exception being correlation based methods, which on the other hand use simpler pairwise gene-gene associations and rarely perform systems analysis. Since an advantage of our method is that we do not explicitly use the interactome and therefore could detect previously undiscovered relationships we think that our approach should mainly be compared to methods that only use transcriptomics. We have therefore added the direct training neural network based approach more discussed in Reply 2.2. Interestingly, both our AE based method and the direct approach could classify patients and controls perfectly, yet our approach was significantly more enriched for GWAS associations. Second, in Fig. 4 we analysed how samples clustered in our deep AE and compared the different layers as well as to a shallow AE. As pointed out by the reviewer this showed good clustering in several of the layers when compared to the naive principal component approach. However, we feel this figure to be somewhat misleading as the Silhouette Index (SI) generally was lowest for the third hidden layer. We therefore recreated the metrics to test the number of times that the corresponding layer were at least 5% above the SI of principal component analysis (Fig. 4a). This metric clearly show that third layered to be most significantly clustered ($P=6.27 \times 10^{-5}$) and to a lesser extent also second layer ($P=1.69 \times 10^{-3}$) and that first layer did not show this clustering ($P=0.125$). This analysis emphasized higher clustering in disease compared to marginal ones, and shows a significantly tighter cell-type clustering compared to PCA. We thank the reviewer for his/her careful assessment of this result and feel that the new figure better displays this fact.

Comment 2.7: *“The RNAseq expression data analysis appears disconnected, and there are few significant associations in Fig. 5. Validation would mean that DAE trained on microarray data, give disease enrichments on RNAseq data, and vice-versa. This would demonstrate that the AE didn’t learn artifacts related to the experimental techniques, but rather real biological (gene) signatures.”*

Reply 2.7: We agree that validation might be a too strong claim. In order to avoid confusion about this section we have now clarified that this section was to show the generality of the suggested approach to multiple measurement types.

Less critical remarks:

Comment 2.8: Sparse AE (SAE) and Denoising AE (DAE) have been shown to work better than simple AE, so I recommend trying those as well. (Also note that their abbreviations unfortunately overlap with those used by the authors for shallow and deep AE).

Reply 2.8: We thank the reviewers for the suggestion of including the denoising and sparse AEs. Analysis of these two show similar behavior between our standard AE and these AEs, i.e. a PPI associations that is strongest in the first hidden layer compared to second and third (see Fig. S6 and S7). Interestingly, the difference between the layers seems smaller for the sparse AE, but of course this might be affected by the sparsity parameter. In the revised manuscript we now also changed notion use the term shallow AE and deep AE to our tested AEs to avoid confusion in the abbreviations.

Comment 2.9: “In the abstract the authors write “Yet biological networks, commonly used to define such modules are incomplete and biased toward[s] some well-studied disease genes”: I think this depends very much on the network. For example, co-expression networks, are hardly biased towards well-studied genes, as RNAseq (and even modern microarrays) cover essentially all transcripts. Even for PPI there are (raw) experimental data with little bias, if any, but of course this is different when looking at integrated network, such as STRING, that may use literature searches etc. to complement and enforce links with additional (human biased) annotation.”

Reply 2.9: We agree that the incompleteness of biological networks might be general and we have therefore specified this to PPI networks. We also agree that the knowledge bias is less of a problem for correlation networks. Although correlations are indications of interactions its likely that those networks are much more incomplete as they are either based on a limited data-set (such the one using raw PPI data) or using the crude inference of an interaction based on correlation. To our knowledge these both incompleteness and knowledge bias are critical problems whenever network analysis is being performed (see e.g. Menche et al in Science 2015 (PMID:25700523). Therefore our methodology that avoids the explicit inference of a physical network, yet discovering functionally related disease-associated genes is of high importance. We have now also commented on some alternatives to STRING exists using co-expression networks and clustering.

Comment 2.10: “3B\$ as a lower limit for new drug might be a bit much. <https://www.policymed.com/2014/12/a-tough-road-cost-to-develop-one-new-drug-is-26-billion> approval-rate-for-drugs-entering-clinical-de.html quotes the out-of-pocket cost of \$1.4 billion, but also notes the time cost (estimated at \$1.2 billion), so it’s not far off ... Maybe give more than one reference and avoid an exact figure?”

Reply 2.10: Yes this numbers are may differ on the exact definition and is not a core part of our manuscript. We therefore decided to remove this first part introductory medical part of the introduction to have more space explaining the logic of using networks and auto-encoders within systems medicine. We feel this more appropriate to the general Nature communication reader.

Comment 2.11: Moreover, individual markers for drug selection generally work poorly, and the choice of drugs in complex diseases is often based on trial and error strategies, causing suffering for patients and increasing costs for health care.”: Note that some individual markers work quite

well: genotyping of the CYP2D6 enzyme is predictive of the processing of many antidepressants and antipsychotic medications, and genotypes in IL28B strongly associate with treatment success of chronic hepatitis C. Also some cancer therapies strongly rely on genotyping nowadays.

Reply 2.11: Again we removed this discussion from our manuscript to focus our manuscript on networks similar to Reply 2.10.

Comment 2.12: “Omics could potentially revolutionize medicine by analyzing disease”: I’d use ‘omics’ as a short of ‘-omics data’, but not for ‘-omics data analysis’.

Reply 2.11: Again we removed this part of the introduction from our manuscript of similar reason as Reply 2.10 and 2.11.

Comment 2.12: “Systems medicine applications to date have often utilized the fact that disease genes are functionally related and their corresponding protein products are highly interconnected and co-localized within networks []”: I think this can only be described as a trend. Also, what’s colocalization here (other than being highly connected)?’.

Reply 2.12: We agree that this statement refers to a trend within systems medicine and we have revised the introductory text accordingly. We also removed the word colocalization since it does not add any new information apart from highly interconnected.

Comment 2.13: “To this end we utilized the protein-protein interaction data in STRING, as a first step to remove the essence of the knowledge of interactome in defining the phenotypic modules.”: Essence of knowledge? Not clear.’.

Reply 2.13: We agree and have revised the sentence to yield: “To this end we utilized closeness and centrality of the protein-protein interaction data in STRING²⁰, as a first step to test if the derived gene-sets was linked to previous disease module research.”

Comment 2.14: “The rationale is that by inducing identity mapping from input to output we can readily inspect the resulting deep representation from a disease module standpoint.”: Reformulate.’.

Reply 2.14: We have reformulated the sentence to yield: “The underlying hypothesis being that an auto-encoder representation represent a maximal unbiased nonlinear compression of the data and therefore closeness with this compressed space would indicate functional similarity similar to protein-interaction modules, which might also well describe upstream disease factors according to the disease module hypothesis[Ref. 2].”

Comment 2.15: “Fig. 1A: First the DAE has 3x as many parameters as the SAE, so the comparison is not straightforward. Still, 3x64 and 3x128 DAE do better than a 512 SAE, so I’d point that out. Better would be to have SAEs with 192, 384, ... nodes. Finally SAE overtake DAE between 512 and 1024 nodes, yet it would be good to know more accurately at what node number.’.

Reply 2.15: We discuss it by the observation that not improvement in R^2 values with increase in hidden nodes. The parameters needed to estimate is not same as referee is suggesting however, while including all the three layer features he is correct. The estimated parameters in 1 layer model with 1024 node is $2*20848*1024$ (encoder/coder weights) + 1024 (biases in the hidden layer) + 20848 (biases for decoder) = $1024*(2*20848 + 1) + 20848 = 42,718,576$, whereas in the case of 3 layer model with 512 node is $20848*512 + 2*512*512 + 20848*512$ (encoder/middle layers and

decoder) + 3*512 (biases in middle layers)+20848 (biases for decoders) = 512*(2*20848 + 2*512 +3) + 20848 = 21895024. Thus the number of parameters to be learned is **1.95 times** many in the shallow 1024 AE compared to our 512 node deep AE. Similarly, if we compare the parameters to train the 512 shallowAE and 512 deepAE, the ratio is **0.97 times**. We have also included a comment about the number of parameters of these models in the revised results.

Comment 2.16: “Fig. 2: More space between A and B!”.

Reply 2.16: We revised Fig. 2 as Fig. 3 in the revised manuscript by making more space between (a) and (b). It should be completed in two weeks after having the three data point results.

Comment 2.17: “Fig. 3: Are the bars representing averages (there is one light-up for each node)? If so, also show the stand deviation (or full distribution of p-values). As there are 512 light-ups giving rise to gene sets, shouldn't there be a correction for multiple hypotheses testing? Also, the light-up of each node results in a set of gene scores at the output layer. What cutoff is used to define discrete gene sets (allowing to users Fisher's exact test)? How about using classical gene enrichment using the score itself (e.g. by testing if the GWAS disease genes tend to fetch higher scores than expected)? Also, make color code for layers consistent with Fig 2.’.

Reply 2.17: This question refers to two different plots, i.e. Fig. 2 and 3 in our previous manuscript which corresponds to Fig. 3 and 4 in the current version. We will refer to the current Figure numbering below. First, Fig. 4 displays the average Silhouette Index across the 512 different nodes and shows a general average clustering. Secondly, Fig. 3 refers to disease enrichment analysis from our reverse training approach, which is a single gene ranking. For simplicity we displayed the enrichment for top 1000 genes in all those cases. Thus, in neither of the cases multiple testing are being performed. Indeed, as indicated by the reviewer the 1000 selection for enrichment was selected somewhat arbitrary, and was chosen to correspond to ~5% of the cases. Ranking based gene set enrichment analysis represent a more flexible method for this, but when comparing sets from different studies we might get the highest enrichment for very different rankings. This may lead to somewhat harder comparisons across the different tests. Nevertheless, as suggested by the reviewer we performed gene set enrichment analysis (GSEA) of our disease rankings from reverse training. These analyses showed similar results as our original analysis and are now included in Fig. S2, i.e. highest and significant enrichment in third layer,.

Comment 2.18: “Methods: “The micro-array data is log transformed normalized values.”: Did you mean “The microarray data were first normalized by a log transformation.”?”.

Reply 2.18: We did log transformation on only RNA-seq data. The micro-array data already normalized and log transformed in the source [Ref. 4]. We revise the beginning sentence of methods section by “The available micro-array data at Ref.⁴ represents normalized log transformed values” also we revise by adding “.....both, micro-array and RNA-seq data,” in the same paragraph.

Comment 2.19: “The normalized expression matrix [...] is input and output signals for training the auto-encoder”: Did you mean “The normalized expression matrix [...] is used both as input and output for training the auto-encoder”?

Reply 2.19: We revise it for more clarity as “The normalized expression matrix $[e_{i,j}]$ is used both as input and output for training the auto-encoder with sigmoid activation function.”

Comment 2.20: “This model has fewer reconstruction errors in comparison with similar hidden node of the one-layer model (SAE)”: Rephrase. (Also see my previous comment on comparing with the same number of nodes per layer, but different total number of nodes.)’.

Reply: We revised the text “The reconstruction errors of this model is lesser degree in comparison with the similar hidden node of the one-layer model (SAE)”. Combine it by previous the reply.

Comment 2.21: “The text following “(2) Training a supervised neural network” is unclear and seems to have mistakes.’.

Reply 2.21: We revised as “Training a supervised neural network on the compressed representations”.

Comment 2.22: “The performance of the predicted genes was demonstrated in terms of Fisher p-value using a hypergeometric test.”: Strictly speaking Fisher’s (exact) test is different from a hypergeometric test: The former converges to the latter for large numbers, but also works exactly for small numbers.’.

Reply 2.22: Yes, we agree with the referee, we replace “Fisher’s (exact) test” with “hypergeometric test”.

REVIEWERS' COMMENTS:

Reviewer #1 (Remarks to the Author):

The authors describe an analysis with a multi-layer autoencoder of gene expression. The authors find that different types of genes are found to influence nodes at different layers, with protein-protein interactions being most manifest at low layers and disease and cell-type signals being captured at higher layers. The authors revisions have addressed the concerns from my previous review. Scientifically, I no longer have concerns.

The presentation of the work could likely be more effective. There are a number of very lengthy paragraphs that could potentially be split into more focused paragraphs. A light restructuring of the paper, retaining the current content, into more focused paragraphs would probably increase its readability.

Reviewer #2 (Remarks to the Author):

The authors did a great job in responding to my comments.

There's just one answer to my comment 2.5, which maybe could still be clarified:

"Third, as suggested by the reviewer, we also used a gene network where interactions are present if two genes are involved in the same disease. Again, although this network has a different meaning we found the strongest association in the first hidden layer."

Maybe I misunderstand, but I thought the hypothesis was that the first layer is the most molecular (i.e. "low level"), while the last layer is the most "integrated", i.e. the closest to the disease. Wouldn't that imply that a disease gene network, as studied here, should not have the strongest association in the first, but in the deeper layers?

Referee 1:

Comment 1.1: "The authors describe an analysis with a multi-layer autoencoder of gene expression. The authors find that different types of genes are found to influence nodes at different layers, with protein-protein interactions being most manifest at low layers and disease and cell-type signals being captured at higher layers. The authors revisions have addressed the concerns from my previous review. Scientifically, I no longer have concerns."

Reply 1.1: We are thankful to the reviewer for finding out the appropriateness of our responses and scientifically accepting our manuscript.

Comment 1.2: "The presentation of the work could likely be more effective. There are a number of very lengthy paragraphs that could potentially be split into more focused paragraphs. A light restructuring of the paper, retaining the current content, into more focused paragraphs would probably increase its readability."

Reply 1.2: We agree with the reviewer. We revised the manuscript by splitting some long paragraph and making them more focused. We believe that this increased the readability of the manuscript.

Referee 2:

Comment 2.1: "The authors did a great job in responding to my comments."

Reply 2.1: We thankful to the reviewer for finding out the suitability in our responses that address his/her concerns raised previously.

Comment 2.2: "Third, as suggested by the reviewer, we also used a gene network where interactions are present if two genes are involved in the same disease. Again, although this network has a different meaning we found the strongest association in the first hidden layer."

Maybe I misunderstand, but I thought the hypothesis was that the first layer is the most molecular (i.e. "low level"), while the last layer is the most "integrated", i.e. the closest to the disease. Wouldn't that imply that a disease gene network, as studied here, should not have the strongest association in the first, but in the deeper layers?"

Reply 2.2 We agree with the reviewer that our overall hypothesis is that closeness between genes that co-localize in the protein-protein interaction (PPI) network is mostly profound within the first layer, and that clustering of patients in different diseases and association to GWAS is coming from deeper layers. However, the validation using the disease-gene network is a hybrid of these two. First, it is very correlated with the PPI network as well as disease relevance. From its definition it is evident that edges does not represent any physical interactions. For example, if gene A and B are connected, as well as B and C, also A and C needs to be connected from the definition. Therefore, we get a highly correlated representation with fully connected cliques. Second, the distance matrix of the two networks are correlated (Pearson correlation coefficient = 0.30 with $P < 2.20 \times 10^{-16}$), as well as the betweenness centrality of nodes from disease network and PPI network show strong correlations to each other (Pearson correlation coefficient = 0.33, $P < 2.20$

$\times 10^{-16}$, and overlap of top 100 most central genes in both is highly significant, Odds Ratio = 15.17, $n=26$, $P < 9.95 \times 10^{-15}$). This means that the disease network is a mix between a network representing molecular function as well as disease associations. We have now clarified this similarity in the revised online supplement.